# Insights into the Scenario of SARS-CoV-2 Infection in Male Reproductive Toxicity

**DOI:** 10.3390/vaccines11030510

**Published:** 2023-02-22

**Authors:** Anirban Goutam Mukherjee, Uddesh Ramesh Wanjari, Abilash Valsala Gopalakrishnan, Sandra Kannampuzha, Reshma Murali, Arunraj Namachivayam, Raja Ganesan, Kaviyarasi Renu, Abhijit Dey, Balachandar Vellingiri, D. S. Prabakaran

**Affiliations:** 1Department of Biomedical Sciences, School of Biosciences and Technology, Vellore Institute of Technology (VIT), Vellore 632014, India; mukherjee1anirban@gmail.com (A.G.M.); uddeshwanjari786@gmail.com (U.R.W.); sandrajacobkannampuzha@gmail.com (S.K.); reshmamurali.rm@gmail.com (R.M.); arunrajnamashivayam97@gmail.com (A.N.); 2Institute for Liver and Digestive Diseases, College of Medicine, Hallym University, Chuncheon 24253, Republic of Korea; vraja.ganesan@gmail.com; 3Centre of Molecular Medicine and Diagnostics (COMManD), Department of Biochemistry, Saveetha Institute of Medical and Technical Sciences, Saveetha Dental College & Hospitals, Saveetha University, Chennai 600077, India; kaviyarasirenu.92@gmail.com; 4Department of Life Sciences, Presidency University, Kolkata 700073, India; abhijit.dbs@presiuniv.ac.in; 5Stem Cell and Regenerative Medicine/Translational Research, Department of Zoology, School of Basic Sciences, Central University of Punjab (CUPB), Bathinda 151401, India; geneticbala@buc.edu.in; 6Department of Radiation Oncology, College of Medicine, Chungbuk National University, Chungdae-ro 1 Seowon-gu, Cheongju 28644, Republic of Korea; 7Department of Biotechnology, Ayya Nadar Janaki Ammal College (Autonomous), Srivilliputhur Main Road, Sivakasi 626124, India

**Keywords:** SARS-CoV-2, male infertility, reproductive toxicity, immunology

## Abstract

COVID-19 has become a significant public health concern that has catastrophic consequences for society. Some preliminary evidence suggests that the male reproductive system may be an infection target for SARS-CoV-2. SARS-CoV-2 may be transmitted sexually, according to preliminary research. Testicular cells exhibit a high level of the angiotensin-converting enzyme 2 (ACE2) receptor, which enhances the entry of the SARS-CoV-2 into host cells. Some instances of COVID-19 have been documented to exhibit hypogonadism during the acute stage. Furthermore, systemic inflammatory reactions triggered by SARS-CoV-2 infection may cause oxidative stress (OS), which has been shown to have profoundly deleterious consequences on testicular functioning. This work gives a clear picture of how COVID-19 may affect male reproductive systems and calls attention to the many unanswered questions about the mechanisms by which this virus can be linked to men’s health and fertility.

## 1. Introduction

The coronavirus disease (COVID-19), caused by SARS-CoV-2, was first identified in Wuhan (China) in December 2019 [1,2] and has since spread fast over the world, resulting in millions of deaths and significant disruptions to healthcare systems in several countries throughout the world. Although the respiratory system is the primary target of the virus, multiple investigations have shown that other tissues, including the kidney, liver, muscles, neurological system, and spleen, are also vulnerable to virus infection [3,4]. The impact of COVID-19 on the male reproductive system is another topic that could garner significant community interest. Some studies portray that SARS-CoV-2 may negatively impact male fertility [2,5,6].

In the testes, high levels of ACE2 receptor protein are expressed on the surface membranes of spermatogonia, Leydig cells, and Sertoli cells [7,8,9,10,11,12]. Additionally, prior research has demonstrated that SARS-CoV can cause consequences such as sterility, orchitis, and injury to the testicles [13]. According to a case report study, an orchiepididymitis diagnosis in a 14-year-old kid recently surfaced to be linked to the COVID-19 virus [14].

Further research is needed to determine the impact of SARS-CoV-2 on the male reproductive system and the likelihood of infertility due to the virus infection. However, it is essential to remember that spermatogenesis is inhibited, and fertility may be affected by any systemic inflammatory disease accompanied by a high body temperature [15].

Despite the immune system’s critical antiviral role [16,17], orchitis can develop if immune factors are produced in excess. After a SARS-CoV-2 infection, the body’s adaptive immune response is a massive release of inflammatory factors and chemokines; however, this can lead to uncontrolled inflammation, leading to a significant decline in lymphocyte numbers and dysfunction in the adaptive immune response resulting in secondary autoimmune orchitis [18,19]. Some of the root causes of these effects can be traced back to the control of the immunological response by sex hormones such as testosterone [18,19,20]. We can better respond to illnesses with a deeper understanding of these epidemiological findings [20]. However, the impact on male reproduction and the underlying causes remain unknown, despite epidemiological studies suggesting that males are at a higher risk of morbidity and mortality associated with COVID-19 [21,22,23,24,25,26]. This work sheds light on the probable pathophysiology, invasion mechanisms, and the immunological background of SARS-CoV-2 in male reproductive toxicity [27].

## 2. Host Cell Invasion Mechanism

The viral S proteins and the host cell receptor ACE2 work together to allow viruses to enter host cells [28,29,30,31]. Transmembrane protease serine 2 (TMPRSS2) on the host cell’s surface prepares the S protein and another cellular protease to split the S protein into S1 and S2 subunits when the S protein interacts with the ACE2 receptor [29,32,33]. Although most infected individuals exhibit COVID-19 symptoms by day five following initial contact, the incubation period for COVID-19 lasts from 1 to 14 days [34,35]. Once a person contracts the virus, SARS-CoV-2 replicates inside the airway epithelial cells. The previously mentioned interaction mediates this between host ACE2 and TMPRSS2 [29]. Fever, myalgia, a painful throat, and shortness of breath are common symptoms of airway epithelium infection [36,37]. Additionally, alternate host cells that express more ACE2 include enterocytes, which might cause COVID-19 gastrointestinal symptoms such as diarrhea [36].

ACE-2 helps convert angiotensin II (Ang II) into a hormone with a less inflammatory effect [36,37,38]. Nuclear factor kappa B (NF-kB) regulates the pulmonary vasoconstriction and inflammation induced by Ang II binding to type 2 angiotensin receptors. This route stimulates the activity of cytokines. Increased pulmonary vein permeability and inflammatory injury to lung tissue result from low ACE-2 levels and high AngII levels [38]. Therefore, a cytokine storm resulting from an uncontrolled immune response to lung damage is assumed to cause lung damage during more severe phases of SARS-CoV-2 infection [39,40]. Ang II is the critical regulator at the outset of this procedure. TMPRSS2 facilitates proteolysis, which is required for viral attachment to the cell surface [29]. Since SARS-CoV-2 coronavirus infection requires binding to the ACE-2 receptor, this protease is essential for viral replication [41]. Androgens are a promotor for the TMPRSS2 gene, and the transcription of this gene is thought to be androgen receptor-dependent [42]. Not only do factors known from research on the renin–angiotensin–aldosterone axis affect Ang II levels, but androgens also have a role [43].

## 3. SARS-CoV-2 Infection and Male Infertility

### 3.1. ACE2 and RAAS

The renin–angiotensin–aldosterone system (RAAS) is a hormonal cascade that maintains homeostasis by controlling arterial pressure and extracellular volume. Angiotensinogen is converted to angiotensin I (Ang I) by the renin–angiotensin system (RAAS). ACE and ACE2 create the bulk of the rest of the pathway’s output, mediating various functions. ACE is a membrane-bound exopeptidase that catalyzes the conversion of Ang I to Ang II by cleaving off the C-terminal dipeptide [44,45]. The gene encoding ACE2 is found on chromosome Xp22, and the protein it encodes is an 80S type-I transmembrane protein. ACE2 has an extracellular catalytic domain and an intracellular tail, just like ACE. The similarity between ACE and ACE2 in the catalytic domain is 42% [46,47]. However, ACE2 does not convert Ang I to Ang II. Thus, its action is not suppressed by ACE inhibitors (ACEIs) [48]. Ang I is converted to Ang II by ACE, while Ang I is converted to angiotensin 1-9 by ACE2 (Ang1-9). ACE2 converts Ang II into the heptapeptide metabolite known as angiotensin 1-7 (Ang1-7). The final two products of this route, Ang II and Ang1-7, exert their effects through the angiotensin type I receptor (AT1R) and the mast cell-activating kinase receptor (MasR), respectively [45,49,50] (Figure 1). The testes, epididymis, and possibly the prostate gland have been shown to have the usual and primary components of RAAS [51,52]. In the testis and epididymis, the RAAS components control steroidogenesis, testosterone synthesis, spermatogenesis, and sperm contractility [51]. In men with immature spermatogenesis, MasR, and ACE 2 mRNA levels were reduced in testicular samples. Furthermore, in males unable to conceive due to non-obstructive azoospermia, neither RAAS subunit was detected in the seminiferous tubules [51].

### 3.2. ACE2 and TMPRSS2

ACE2 is located in many parts of the human body, including the lungs, kidneys, bladder, skin, lymph nodes, liver, and gallbladder [47,53,54]. The testes were found to have the most significant ACE2 expression in the male reproductive system [53]. Spermatogonia, Leydig, and Sertoli cells express ACE2 at high levels, while spermatids and spermatocytes hardly present any detectable expression [55]. The RAAS subsystems appear to be crucial for male fertility. ACE2 controls testosterone synthesis and steroidogenesis by converting Ang II to Ang1–7 in Leydig cells [56]. Second, ACE2 is known to control the microvasculature of the testicles, which maintains a healthy interstitial fluid volume [57], indicating its importance. Younger males have more ACE2 expression in their testes than older men [8]. In humans, ACE2 expression begins during puberty and reaches its highest level to maintain male fertility [58]. Fertile males have been found to have higher ACE2/Ang1–7/MasR expression levels than infertile men [51]. Multiple cellular processes, including epithelial sodium homeostasis, microvesicular endothelial cell angiogenesis, and tubulogenesis, are regulated by TMPRSS2 [55,59,60]. TMPRSS2 was discovered in prostasomes. It was hypothesized that TMPRSS2 in prostasomes is necessary for normal sperm production [50,61]. In the male reproductive system, ACE2 is involved in regulating steroidogenesis. There is also evidence that ACE2 receptors contribute to fertility. One study found that infertile men with substantial spermatogenesis dysfunction had lower levels of ACE2 than fertile participants [51]. The expression of TMPRSS2 is greater in men than in women. Because the androgen response element acts as a transcriptional promotor for TMPRSS2, this is likely to be the case. Co-expression of ACE2 and TMPRSS2 is hypothesized to be necessary for viral entrance into cells [62,63].

### 3.3. SARS-CoV-2 in Erectile Dysfunction

Endothelial dysfunction induced by COVID-19 may occur systemically, including in the penile erectile tissue, which contains many blood vessels lined with endothelium [64,65]. Pilot research describing the histological characteristics of penile tissue in subjects who healed from symptomatic COVID-19 infection and acquired significant erectile dysfunction. Patients with severe erectile dysfunction who underwent penile prosthesis surgery provided their penile tissue for this study [65]. Two males with known COVID-19 infection histories and two without such histories provided samples. The findings of this study are the first to show that the COVID-19 virus can persist in the human penis for a long time after infection [65]. This study adds to the growing evidence linking COVID-19 infection and erectile dysfunction by suggesting that extensive endothelial cell dysfunction may play a role [65]. Another study examined the incidence of erectile dysfunction among participants with a recorded diagnosis of COVID-19 and the relationship between COVID-19 and erectile dysfunction. This study analyzed data from the Sex@COVID online survey to obtain a sample of sexually active Italian males with SARS-CoV-2 infection. This study found preliminary evidence in a real-world population that erectile dysfunction is a risk factor for acquiring COVID-19 and may result from COVID-19 [66]. Vascular impairment and the development of more severe erectile dysfunction may result from immunothrombosis affecting the penile vessels and causing endothelial dysfunction. Even after the conclusion of the acute phase, erectile dysfunction may persist due to other cardiovascular consequences of COVID-19, such as cardiomyopathy and myocarditis, which may progress into long-term cardiovascular illness. Impairment of erectile function in COVID-19 patients might also be due to other factors, such as pulmonary fibrosis resulting in hypoxia in the penile vascular bed [2,66], a symptom of COVID-19 that might negatively affect sexual health. In this sense, the study by Kresch et al. was the first to show that the COVID-19 virus persists in the penis for a considerable time after the first infection in people. Extracellular virus particles were found close to penile vascular endothelial cells in patients who tested positive for COVID-19 using transmission electron microscopy (TEM). Notably, tissue samples collected from males tested negative for COVID-19 contained no detectable viral particles. The expression of the eNOS gene was lower in the corpora cavernosa of COVID-19-positive males than in COVID-19-negative men, as shown by immunohistochemistry [5,65,66].

### 3.4. SARS-CoV-2 and Oxidative Stress

Inducing OS through the activation of inflammatory responses is thought to be a typical pathogenic strategy used by SARS-CoV-2 to impair numerous physiological systems by causing oxidative damage to host tissues [67]. Recent research looked at how males with COVID-19 fared on measures of ACE-2 enzymatic activity in their seminal fluid, along with inflammatory and anti-inflammatory cytokines, oxidative stress, apoptotic factors, and semen quality [67,68,69]. Significant changes in the volume, motility, morphology, concentration and number of spermatozoa occur alongside the presence of OS markers in sperm cells, which are higher in affected individuals than in healthy controls [68]. More research is needed to determine the effect of OS on the seminal fluid of COVID-19 patients over a more extended period because the duration of follow-up for these changes was only 60 days, which is less than one spermatogenesis cycle (at least 74 days). After entering the body, the virus damages respiratory tissues and triggers inflammation, ocular toxicity, sperm destruction, and death. Dietary supplements for fertility, which work naturally by counteracting the action of free radicals, may restore the oxidative equilibrium and provide relief from this problem. Male infertility caused by OS has been studied extensively because of its effects on sperm quality, disruption of sperm function, and morphology [70,71]. Overproduction of ROS during SARS-CoV-2 infection can primarily activate NF-kB receptor-like pathways. The result is an uptick in inflammatory reactions, triggering even more cytokine production [72]. Innate immune action of receptor DPP4/CD26 may also contribute to cytokine storm triggered by ROS generation. [73,74,75]. OS caused by SARS-CoV results in a significant generation of macrophage-derived oxidized phospholipids, stimulating cytokine oversupply and increasing the host inflammatory reaction via oxidant-sensitive inflammatory processes. NF-kβ is centrally involved in SARS-CoV infections [76,77]. It possesses binding sites in the promoter regions of genes associated with apoptotic elements and pro-inflammatory mediators. SARS-CoV 3CLpro (a viral protease) significantly increases ROS generation, which triggers NF-kβ-mediated cell death. SARS-CoV works via the MAPK pathway, which may trigger mitochondrial apoptotic pathways via Bax oligomerization [67,76,77]. In response to OS, the increase in inflammatory genes in peripheral blood mononuclear cells (PBMC) of SARS-CoV-infected patients implies that SARS-CoV infection initiates the vicious cycle of inflammation and OS [67,76,77].

### 3.5. SARS-CoV-2 in Vasculitis

The endothelial cells lining veins and arteries express ACE-2. Endotheliitis, systemic vasculitis, and disseminated intravascular coagulation (DIC) [78] are all possible outcomes of SARS-CoV-2 infection of these cells. Thrombosis and thromboembolism have been high in people with COVID-19 [79], which may be one possible explanation. In conclusion, a vasculitis/endothelitis related to SARS-CoV-2 infection may underlie the orchitis-like illness experienced by some individuals with COVID-19 [80]. The presence of inflammatory cells in the interstitial cells of testicular tissues and elevated seminal levels of IL-6, TNF-α, and MCP-1 was also explored [5,81].

### 3.6. SARS-CoV-2 in Azoospermia and Cryptozoospermia

Non-obstructive azoospermic (NOA) or cryptozoospermic (CZS) patients undergoing medical treatment to maintain or enhance spermatogenesis are likely to be the most at-risk group of male infertility patients during the SARS-CoV-2 pandemic (Figure 2). The patient with hypogonadotropic hypogonadism (HH) is a good illustration; HH causes azoospermia due to insufficient activation of the testes by pituitary gonadotropins [82]. Although there is currently no cure for the illness, medicinal therapy has been investigated to optimize or induce spermatogenesis, thus increasing the likelihood of spermatozoa being recovered surgically or ejaculated [83,84,85,86]. In order to maintain fertility and enable future ART, sperm banking is recommended for patients with NOA who react to medical therapy. There may be a limited time frame for sperm cryopreservation in patients who develop CZS or severe oligozoospermia due to treatment [87]. If sperm banking is not completed, patients may undergo surgical sperm retrieval, which can be a clinical and financial hardship [88]. Heat stress from a varicocele can harm spermatogonia of type B, pachytene spermatocytes, and early spermatids [89,90,91,92,93,94].

### 3.7. SARS-CoV-2 Infection, Testis Damage, and the Associated Immunological Scenario

Studies have observed that SARS-CoV-2 goes to a cell with the help of the ACE2 receptor, which was present predominantly in the epithelial lung cells [31,95]. ACE2-mediated SARS-CoV-2 invasion might result in viral infection, which could harm testicular tissues [96,97,98,99,100].

Although, few studies could not find the presence of SARS-CoV-2 in the testes of the infected individuals [97]. More advanced and increased sample sizes are required to confirm this study’s results further. Male patients with COVID-19 should have their reproductive function monitored and assessed, especially in young males. In addition, men tend to show worse clinical problems when compared to women and have been linked to a greater probability of severe complications [101]. Recent findings reveal that SARS-CoV-2-infected cells express more of the autophagy receptor SQSTM1/p62, which suggests a reduction in autophagic flux. It could suggest that the degree of autophagy may be restricted by SARS-CoV-2, eventually affecting male reproduction [100]. Another research finding implied that SARS-CoV-2 might contribute to the underlying pathophysiology of viral orchitis and the subsequent testicular injury by inducing a secondary autoimmune response [64].

Immunohistochemical examination revealed significantly greater numbers of CD^3+^ T cells and CD^68+^ macrophages in the testis of patients affected with the SARS-CoV-2 compared to the normal healthy samples [102]. Because the RAAS is active in the testis, SARS-CoV-2 infiltration and disruption of testicular ACE2 may result in infection, loss of anti-inflammatory effects, coagulation, and inflammation, which may interfere with spermatogenesis and testicular function. The testis’ most common and abundant immune cells are the testicular macrophages, which are phagocytic toward cell waste, foreign objects, and invasive pathogens. They support maintaining the immune-privileged condition inside the testis and multiply during inflammation. Some other vital cells include the epithelial cells involved in the renewal of the spermatogonial stem cells via various immunological factors, including insulin-like growth factor binding protein 2, glial-derived neurotrophic factor, stromal cell-derived factor 1, macrophage inflammatory protein 2, and fibroblast growth factors [103]. Spermatogonial stem cells and spermatogonia are susceptible to SARS-CoV-2 infection because they express ACE2 and TMPRSS2. In a study conducted by R. Clayton Edenfield et al., the semen samples taken from 23 individuals with moderate or typical instances of COVID-19 revealed that 60.9% of them had elevated seminal leukocytes and 39.1% had oligozoospermia. In addition, the patients’ sperm concentration was lower, and their levels of IL-6, TNF, and MCP1 were higher than those of the males in the control group [104]. Other studies have also reported a similar increase in interleukin levels, suggesting the role of immune response in COVID-affected patients [104]. IFN-γ and IL-2 were reported in high levels in COVID-19 patients and were also found to inhibit sperm motility and viability by promoting lipid peroxidation of human spermatozoa and increasing spermatozoa DNA fragmentation [105]. These results imply that the virus may indirectly influence the hypothalamic–pituitary–testicular axis [45,106].

#### 3.7.1. SARS-CoV-2 Infection and the Anti-Sperm Antibodies

The blood–testis barrier may be compromised by SARS-CoV-2, which might enhance the development of autoantibodies. Given that the testis is abundant with the presence of ACE2 receptors and that they are susceptible to SARS-CoV-2 invasion, changes in semen analysis, changes in the balance of sex hormones, and, most importantly, the formation of anti-sperm antibodies and sperm DNA fragmentation are thought to play a significant role in male infertility [107]. Strong relationships were found between sperm abnormalities and SARS-CoV-2 IgG antibody titers against the spike 1 and the spike 1 receptor-binding region. In the same study, three men were found to have higher levels of anti-sperm antibodies [108].

#### 3.7.2. Role of Androgens on SARS-CoV-2 Infection

Androgens help increase TMPSS2 enzymes, and this gene encodes the transmembrane protease [29,109,110]. Androgen deprivation therapy (ADT) is used to treat prostate cancer. There is a mutual relationship between the androgens and the TMPSS2 gene. Testosterone and the gene expression TMPSS2 in COVID-infected persons have changed the tendency of host cells and proteins combined with the receptor ACE2 [111,112]. The AR gene encodes androgen receptors (Xq11–12). Variations in this gene have been linked to divergent androgen responsiveness. These modifications raise the risk of androgen-related diseases such as prostate cancer and androgenic alopecia (AGA) [113,114]. AGA and androgen changes were interlinked with COVID-19. In some European countries such as Italy and Germany, patients affected with COVID-19 subjects showed lower testosterone patients admitted to ICU. This research outcome shows a link between the androgen levels in the blood and androgen sensitivity [115]. From some point of view, a factor causes SARS-CoV-2 inflammation. This inflammation includes higher levels of cytokines and depletion in the androgen levels in both old and young males [107,116,117].

##### Risks of Higher Androgen Levels

These variations in androgen levels have been seen in the patients treated with ADT in prostate cancer treatment [111,112,118,119]. According to statistics from around the world, males are more likely to contract SARS-CoV-2 coronavirus infections than females. Prepubescent children may also be carriers of the disease, but their symptoms are typically mild [101,120,121,122]. AGA is caused due to the higher androgen levels in the hair follicles as it results in the formation of dihydrotestosterone. In young individuals, severe steroid intake causes hyperandrogenization [118,123]. Adrenal androgens have not yet been considered when studying how androgens affect SARS-CoV-2 infections because they make up a small portion of all androgens and can change throughout the illness, particularly when dexamethasone is administered [124]. At a 7-month follow-up, more than 50% of men who healed from SARS-CoV-2 still had circulating testosterone production that was suggestive of hypogonadism, even though total testosterone levels have risen over time after COVID-19 [125].

##### Risks of Low Androgen Levels

Observations regarding the progression of the illness and its mortality do not support hypotheses regarding the risk of higher androgen levels in the course of SARS. On the other hand, low levels of free or total testosterone were listed as a risk. The less favorable course of the disease was observed in patients who were deficient in androgens and had low testosterone below 5 nmol/L, in those who required testosterone replacement therapy, and in patients who also frequently had low testosterone levels due to COVID-19 pneumonia or other lung conditions [126,127]. Low testosterone levels are thought to be indicative of hormonal changes in men, particularly older men, at risk for poor prognosis or death [128]. Reduced estrogen levels in critically ill men impair endothelial cell function, cause immune response issues, lessen the body’s capacity to eliminate the virus and increase systemic inflammation. Low testosterone may be predictive of a COVID-19 adverse outcome. A clinical study was carried out on 31 male patients who recovered from SARS-CoV-2 and had low levels of testosterone after the treatment of pneumonia [129]. Overall, the survey on males shows lower baselines of free and total testosterone levels. Other authors also discovered significantly lower testosterone, higher LH, and prolactin levels in COVID-19 pneumonia patients [11].

#### 3.7.3. SARS-CoV-2 Infection and the Cytokines Stimulation

When a viral or bacterial infection spreads throughout the body, it causes hypercytokinemia, an uncontrolled hyperinflammatory reaction. Increased cytokine levels harm multiple organ systems because they cause cellular and metabolic dysregulation. Early in hypercytokinemia, levels of chemotactic cytokines (IL-8 and MCP-1) and acute response cytokines (TNF and IL-1) rise, which helps IL-6 levels rise steadily. To maintain the inflammatory processes, IL-6 binds to either the membrane-bound IL-6 receptor (mIL-6R) or the soluble IL-6 receptor (IL-6R), forming a complex that acts on gp130 and controls the levels of IL-6, MCP1, and GM-CSF via the Janus kinase–signal transducer and activator of transcription (JAK-STAT) pathway [130]. The acute phase response that IL-6 and other pleiotropic cytokines trigger increases serum ferritin, complement, CRP, and pro-coagulant factors, many of which can be detected using readily available blood tests. The cytokine storm’s acute phase response is generally exaggerated. Low levels of cytotoxic T cells may contribute to decreased viral clearance because high serum levels of cytokines are inversely correlated with the total lymphocyte count [131]. The cytokine storm may be treated by inhibiting upstream processes involved in or at the level of the cytokine response, such as macrophage JAK-STAT signaling that downregulates the production of IL-1 and IL-6. The time to the therapeutic effect of anti-B lymphocytes directed therapies, such as rituximab, may be too long for them to be clinically relevant. Cell-based target strategies may also be taken into consideration. Targeting the events upstream may therefore be relatively more successful [131]. However, the inflammatory response causes the lymphocytes fighting the SARS-CoV-2 disease to be destroyed. The disruption of the pulmonary epithelial barrier initiates the chain reaction of damage by cytokine storms [132,133].

## 4. SARS-CoV-2 and Its Impact on the Brain Leading to Infertility

Through its impact on the central nervous system, SARS-CoV-2 is hypothesized to affect fertility [134] indirectly. Gonadotropins and sex steroid hormones act to synchronize the brain and testes physiologically. SARS-CoV-2 has been found in the brain, and it has been hypothesized that the virus can infect brain cells because glial cells and neurons express ACE2, making them potential targets for SARS-CoV-2 infection [5,6]. Gonadotropin-releasing hormone (GnRH) is pulsatilely released from the hypothalamus by the hypothalamic–pituitary–gonadal axis (HPG). As GnRH is affected by central nervous system damage, FSH and LH are also affected, affecting the production of sperm [135]. Although there is limited proof that COVID-19 patients’ testicular biopsies and semen samples included viral particles, testicular disease brought on by sudden inflammatory reactions and hyperthermia is becoming increasingly apparent. The altered gonadotropin secretion in COVID-19 is linked to a decreased level of testosterone synthesis [108]. In the H7N9 influenza epidemic, low testosterone has been linked to increased mortality risk, and it has also been linked to increased mortality in COVID-19 [16,136,137,138].

## 5. SARS-CoV-2 and Its Impact on the Prostate Leading to Infertility

One of the primary seminal components, prostate fluid, is secreted by the prostate gland, and the gland’s muscles aid in ejaculation by forcing seminal fluid through the urethra [135]. Around one-third of the total volume of semen is made up of prostate-excreted fluid, which also contains calcium and citric acid [14]. The widespread expression of ACE2 sparked the idea that SARS-CoV-2 might adversely impact male fertility in several testes cells (spermatogonia, Leydig cells, and Sertoli cells) as in some other male genital tract organs such as the prostate gland and epididymis [52]. Since typical, healthy prostate hyperplasia is known to express ACE2, the prostate is more prone to getting SARS-CoV-2 infection, which could impact its secretions [8]. The human prostate’s epithelium highly expresses TMPRSS2. Given that ACE2 and TMPRSS2 are found in the male genital tract, including the testes, male fertility may be harmed by viral invasion, and the prostate is more susceptible to SARS-CoV-2 infection, which could impact its secretions [52,139,140] (Table 1).

## 6. SARS-CoV-2 and the Hormonal Changes Leading to Infertility

The hypothalamic–pituitary–testicular (HPT) axis may be suppressed by an acute severe inflammatory situation, such as in the case of COVID-19, which results in lower levels of luteinizing hormone (LH), follicle-stimulating hormone (FSH), and testosterone. However, a study conducted on 81 COVID-19 male patients showed lower serum levels of testosterone, higher levels of LH, and a lower T: LH ratio compared to age-matched control participants. These results do not support the theoretical perspective [152]. These findings might indicate that testicular cells directly experience the effects of SARS-CoV-2 infection rather than indirectly through the HPT axis. As a result, significant gaps exist in understanding the relationships between male sex hormone regulation and SARS-CoV-2 infection, which requires further investigation [153]. Lower levels of testosterone, lower ratios of testosterone to LH, lower ratios of FSH to LH, and higher levels of LH have all been seen in COVID-19 patients, indicative of primary testicular damage, particularly damage to Leydig cells. The testosterone/LH ratio has also been linked to higher levels of C-reactive protein (CRP), whose rise is correlated with the severity of COVID-19. In COVID-19 individuals, lower testosterone levels are also linked to erectile dysfunction [154,155]. According to the research, ACE2-mediated pathways may be involved in how COVID-19 may affect male infertility, and infertile men may be more vulnerable to COVID-19 [8,64].

## 7. SARS-CoV-2, Semen Contamination, and Transmission

The testicular expression of ACE and TMPRSS2 in the genital tract of men allows speculation for the involvement of the testis during infections and to overcome BTB [156,157,158]. TMPRSS2 and ACE2 mediate the majority of viral spike protein priming. High ACE2 protein expressions are found in the testes [159], increasing the possibility that SARS-CoV-2 infection could threaten male fertility [7,153]. The study by Zhao et al. reported SARS-CoV in testicular epithelial and Leydig cells [13,160]. In other words, it is difficult to prove the existence of SARS-CoV-2 in individuals who have been infected severely since there is a lack of reliable data from large-scale cohort studies. A detailed study is needed to investigate the infectivity and pathogenesis of SARS-CoV-2 and its transmission to understand the implications in both clinical and epidemiology [161].

## 8. Discussions and Future Perspectives

Some of the most commonly used drugs for treating COVID-19 are steroidal drugs, antiviral drugs, and antibiotics [162]. The potential reproductive harm of antiviral medications is frequently overlooked during COVID-19 therapy. Some of the most common antiviral drugs are ribavirin, remdesivir, interferon, ivermectin, lopinavir, chloroquine, and ritonavir [63,163]. Ribavirin, an antiviral drug, was found to impair testosterone levels and dysregulated spermatogenesis [164]. In a clinical study in rats, ribavirin was found to induce sperm abnormalities. This hazardous impact, however, was reversible [164,165]. Clinical trials revealed that ribavirin paired with interferon therapy caused male infertility by lowering sperm count [165,166]. Comprehensive and adequately powered randomized controlled trials examining the effects of Remdesivir in hospitalized COVID-19 patients are scarce. There is insufficient evidence to support the conclusion that Remdesivir is an effective treatment for COVID-19 [167]. Ritonavir has been shown to cause oxidative damage in the testis of rats, resulting in alterations in sperm characteristics and the animals’ antioxidant state [168]. Glucocorticoids have been linked to germ cell death via receptors on germ cells. As a result, glucocorticoids are only indicated for a limited time in treating COVID-19 individuals with rapid progression of imaging symptoms, progressive worsening of oxygenation markers, and significant activation of inflammatory responses in the body [169]. A further study showed that administering therapeutic doses of penicillin G to rats for eight days led to a halt in sperm production [170,171]. More research is needed to understand the role of different biomolecules against male infertility followed by COVID-19 infection. Notably, we cannot confidently claim that the unique SARS-CoV-2 directly infects the testes. However, it is possible, as with other viral infections, that this virus might wreak havoc on reproductive tissues, severely impacting the sexual health and fertility of males. Conversely, any other viral condition can temporarily diminish sperm count for several weeks or months. This makes it challenging to determine whether the decrease in sperm seen in the evaluated research is directly attributable to COVID-19. Regrettably, in most investigations, a semen study was not performed prior to SARS-CoV-2 infection; hence, the semen changes reported in COVID-19 patients may have been present prior to SARS-CoV-2 infection [5]. Future studies on ACE2 and its effects on male genital tissues will necessitate investigation. Altogether, the results presented in this study suggested that, due to its high ACE2 expression, the testis is the organ in the male reproductive system that is most susceptible to SARS-CoV-2 infection. The role of the ACE2 receptors in spermatogonia, seminiferous tubules, Sertoli, Leydig, and prostate epithelial cells is uncertain [27]. Different natural compound sources have been found beneficial in treating SARS-CoV-2 infection in male disorders [171]. Table 2 summarizes the same. Although many initial investigations have been conducted on the antiviral effects of natural substances and Chinese medicine in this field, further in vivo studies using appropriate animal models are required to dissect the fundamental cellular and molecular mechanisms.

## 9. Conclusions

Although progression and sperm morphology are not connected to vaccination, the severity of COVID-19 significantly affects these factors, which can negatively affect fertility. Researchers have postulated and examined potential pathologies in various studies to show how this virus can degrade semen and induce male infertility. Since the beginning of the epidemic more than a year ago, there have been several reports of semen impairment. In order to know whether this impairment is transitory or permanent and to understand how SARS-CoV-2 enters the testis and affects the semen, more research is needed. According to the results of these investigations, SARS-CoV-2 infection may affect male fertility. We suggest that doctors check for COVID-19 after an infection, especially in men of reproductive age who may be experiencing infertility. Molecular biology research on the methods by which SARS-CoV-2 infects host cells and clinical observations raise the possibility that sex hormones, androgens, in particular, contribute to the progression of this infection. Either low or high amounts of testosterone can alter its course. The role of the sex hormone is a double-edged sword since research reveals associations between high and low levels of testosterone in illness progression. As of yet, there is little indication that coronavirus infections can negatively influence male fertility, and it is unclear which illness phases, if any, determine testosterone’s function in the disease’s progression. The effects of SARS-CoV-2 on men’s reproductive health, both immediately and in the long run, are still mostly unknown. Recent research indicates that the possibility of SARS-CoV-2 transmission via seminal fluid is extremely low. A key factor for viral entrance is the coexpression of both ACE2 and TMPRSS2 at appropriate levels, despite the many studies on viral tropism for the male reproductive system. The effects of SARS-CoV-2 infection on male reproductive hormones and semen characteristics are poorly understood. Hormone profiles and semen analyses from cured men gathered over time, as well as larger-scale community-based evaluations for SARS-CoV-2 in semen samples, are required to better understand this virus’s prevalence.

## Figures and Tables

**Figure 1 vaccines-11-00510-f001:**
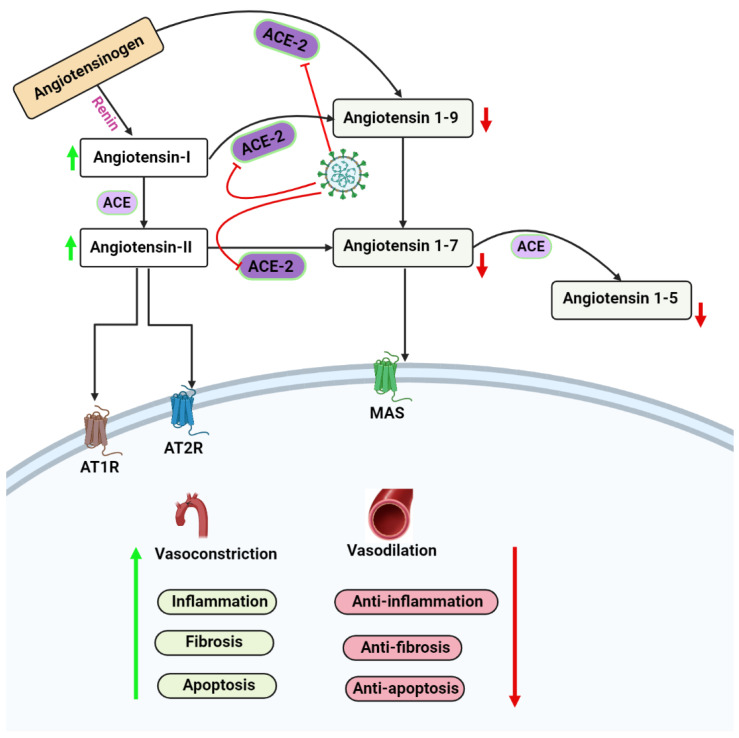
This figure gives an overview of the RAAS system pathway.

**Figure 2 vaccines-11-00510-f002:**
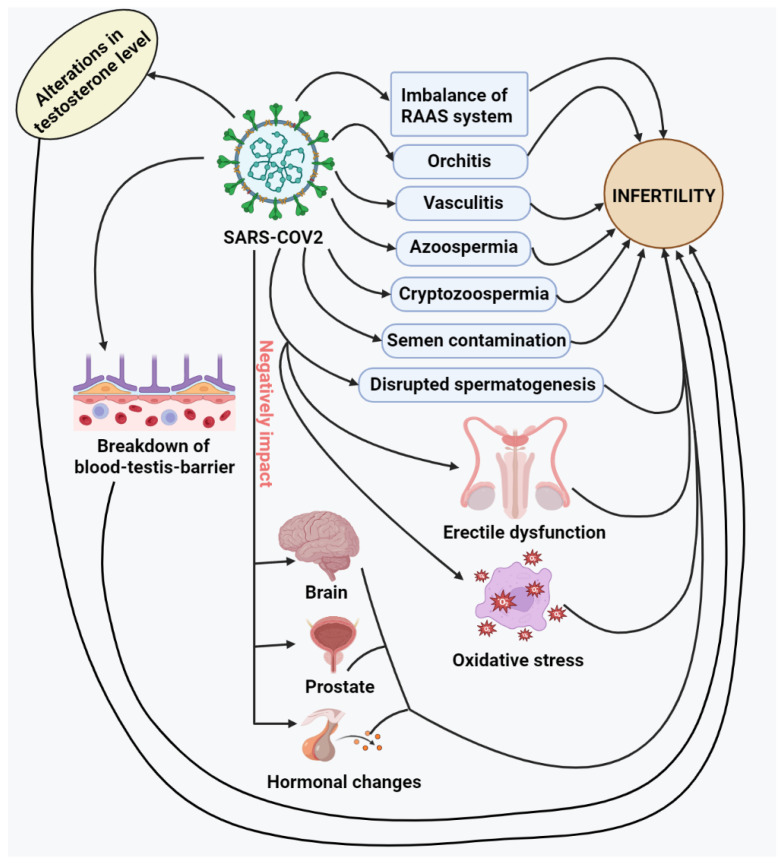
This figure demonstrates the action of SARS-CoV-2 on various organs such as the brain, prostate, and other organs leading to infertility. This diagram also describes that SARS-CoV-2 results in an imbalance of the RAAS system, orchitis, vasculitis, azoospermia, cryptozoospermia, semen contamination, disrupted spermatogenesis, erectile dysfunction, and oxidative stress, all culminating into infertility.

**Table 1 vaccines-11-00510-t001:** Effects of SARS-CoV-2 on various male reproductive parameters leading to male infertility.

Common Name	Scientific Name	Study Type	Study Model	Dose	Effects	Mechanism	Reference
Garlic	*Allium sativum*	In vivo	Adult male albino rats of the Sprague–Dawley strain	30 g	Modify spermatogenesis; spermicidal effect	↓ Plasma and intratesticular testosterone concentration	[141]
Ginger	*Zingiber officinale*	In vivo	Japanese quail (*Coturnix coturnix japonica*)	50, 100, and 150 μL/kg bw	↓ Reproductive cell lipid peroxidation, ↑ fertility	↓ Impairment in sperm membrane and DNA	[142]
Grape	*---------*	In vivo	Wistar rats	2 mg/kg bw	↑ Leydig cell percentage	↑ FSH, LH, and testosterone production; prevent the spermatogenic disruption	[143]
Green tea	*Camellia sinensis*	In vivo	Adult male mice	150 mg/kg bw	↓ Apoptosis, testicular tissue changes, MDA; improve sperm parameters	↓ Free radicals and expression of caspase-3;↑ testosterone levels	[144]
Micro-algae and algae	*Atrhrospira platensis*	In vivo	Barki ram-lambs	---------	↑ Total sperm output and total motile sperm	↑ Anti-oxidative status and semen quality	[145]
Propolis	*---------*	In vivo	Adult male Sprague Dawley rats	50 mg/kg	↓ MDA, GSSG levels; ↑ GSH, ATP levels	↑ Sperm count, motility, and validity; ↓ sperm abnormality	[146]
**Herbal drugs**							
Velvet bean	*Mucuna pruriens*	In vivo	Human	5 g/d	Regulates steroidogenesis; improves semen quality	↓ FSH, prolactin; T, improves LH, dopamine, adrenaline, and noradrenaline levels	[147]
Ashwagandha	*Withania somnifera*	In vivo	Humans	5 g/d	Inhibit lipid peroxidation, ↑ sperm motility, and sperm count; regulate reproductive hormone levels	Oxidative; non-oxidative mechanism	[148]
Amaryllidaceae	*Curculigo orchioides*	In vivo	Male Albino rats	100 mg/kg	↑ Spermatocyte, spermatids	Antioxidative property; ↑ spermatogenesis	[149]
Hemp	*Cannabis sativa*	In vivo	male albino rats	2 mg/kg	Gonadotoxic effect	↓ Sperm parameters, total antioxidant capacity; ↑ ROS	[150]
Tongkat ali	*Eurycoma longifolia* Jack	In vivo	Humans	200 mg	↑ Semen volumes, sperm concentrations, sperm motility, the percentage of normal sperm morphology	↑ Testosterone levels; antioxidant property	[151]

↓ indicates a decline and ↑ indicates an increase or improvement.

**Table 2 vaccines-11-00510-t002:** The potential role of natural compounds in benefiting male fertility after COVID-19 infection.

Sources	Scientific Name	Study Type	Study Model	Dose	Effects	Mechanism	Reference
Garlic	*Allium sativum*	In vivo	Adult male albino rats of the Sprague–Dawley strain	30 g	Modify spermatogenesis; spermicidal effect	↓ Plasma and intratesticular testosterone concentration	[141]
Ginger	*Zingiber officinale*	In vivo	Japanese quail (*Coturnix coturnix japonica*)	50, 100, and 150 μL/kg bw	↓ Reproductive cell lipid peroxidation, ↑ fertility	↓ Impairment in sperm membrane and DNA	[142]
Grape	*---------*	In vivo	Wistar rats	2 mg/kg bw	↑ Leydig cell percentage	↑ FSH, LH, and testosterone production; prevent the spermatogenic disruption	[143]
Green tea	*Camellia sinensis*	In vivo	Adult male mice	150 mg/kg bw	↓ Apoptosis, testicular tissue changes, MDA; improve sperm parameters	↓ Free radicals and expression of caspase-3; ↑ testosterone levels	[144]
Micro-algae and algae	*Atrhrospira platensis*	In vivo	Barki ram-lambs	---------	↑ Total sperm output and total motile sperm	↑ Anti-oxidative status and semen quality	[145]
Propolis	*---------*	In vivo	Adult male Sprague Dawley rats	50 mg/kg	↓ MDA, GSSG levels; ↑ GSH, ATP levels	↑ Sperm count, motility, and validity; ↓ sperm abnormality	[146]

↓ indicates a decline and ↑ indicates an increase or improvement.

## Data Availability

Data are available from the authors on request (A.V.G.).

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
