# Peer review of "Insights into the Scenario of SARS-CoV-2 Infection in Male Reproductive Toxicity"

_vaccines, 2023, doi:10.3390/vaccines11030510_

Round 1

Reviewer 1 Report

At papier  a lot of basic information about the mechanism of COVI-2 infection. Unnecessary in this work. At papier, suggestions rather than evidence. Repeated information about the presence of receptors angiotensin-converting enzyme 2 (ACE2) receptor in at least a few places The work should be shortened to provide real facts and directions of research, especially since the virus changes

Author Response

Reviewer 1

At papier  a lot of basic information about the mechanism of COVI-2 infection. Unnecessary in this work. At papier, suggestions rather than evidence. Repeated information about the presence of receptors angiotensin-converting enzyme 2 (ACE2) receptor in at least a few places The work should be shortened to provide real facts and directions of research, especially since the virus changes

Response: We are very much thankful to the reviewer for reviewing this manuscript and providing such valuable comments to improve the quality of the manuscript. We have again carefully revised the entire manuscript and have tried to address all the issues very carefully.

Reviewer 2 Report

In this manuscript, " Insights into the scenario of SARS-CoV-2 infection in male reproductive toxicity", the authors highlighted the pathophysiology, invasion mechanisms, and immunological background of SARS-CoV-2 in male reproductive toxicity. The only major issue in the manuscript is few statements are overstated.

Comments:

1) In the abstract, there are too many general statements. Either provide a reference or remove them from the abstract. The abstract should be crisp and to the point.

2) There are typing errors in the manuscript that need to be corrected.

3) The authors have discussed COVID-19 and its effects on the male reproductive system in the introduction section. However, there is no direct report available that shows that COVID-19 impacts the male reproductive system, and the ones that are available and authors have cited are indicative only. The authors should tone down the potential claims while writing the reviews in this section. Also, add more references to support your claims.

4) In section 3.1.3, two reports are cited that showed the role of SARS-CoV-2 in erectile dysfunction (53,55). Please describe the nature of the studies and the number of samples used in these papers. Also, rephrase line 181.

5) In section 3.1.4, The role of oxidative stress is described in COVID-19 and its implications for male fertility. Oxidative stress is general in numerous infectious diseases. Please highlight the difference in covid 19 and other conditions that lead to male infertility.

6) Lines 186, 189, 257, 288, 291, 318, and 380 lack referencing. Please recheck all your references and update them with relevant studies (e.g., 91). Please cite only relevant papers and remove irrelevant references.

7) Remove line 336. If testosterone makes men more susceptible, young men should get more infections than older men. There are various factors that lead to the prognosis of COVID-19. For example, is it not common to have low testosterone in a diseased person due to a lack of an active lifestyle and other factors?

8) Rephrase line 370.

9) Please improve the discussion by complementing/supporting more profound and recent literature searches. 

10) The manuscript could benefit from discussing the challenges and future prospects of the field. (optional)

11) Discuss natural products and medicinal compounds that could benefit male fertility after COVID-19. A tabular representational would be better for the readers.

12) Include information in the discussion about preventive biomolecules available. (optional)

 Patel SKS, Lee JK, Kalia VC. Deploying Biomolecules as Anti-COVID-19 Agents. Indian J Microbiol. 2020 Sep;60(3):263-268. doi: 10.1007/s12088-020-00893-4. Epub 2020 Jun 9. PMID: 32647390; PMCID: PMC7282542.

Author Response

Reviewer 2

In this manuscript, " Insights into the scenario of SARS-CoV-2 infection in male reproductive toxicity", the authors highlighted the pathophysiology, invasion mechanisms, and immunological background of SARS-CoV-2 in male reproductive toxicity. The only major issue in the manuscript is few statements are overstated.

Response: We are very much thankful to the reviewer for reviewing this manuscript and providing such valuable comments to improve the quality of the manuscript. We have again carefully revised the entire manuscript and have tried to address all the issues very carefully.

Comments:

1) In the abstract, there are too many general statements. Either provide a reference or remove them from the abstract. The abstract should be crisp and to the point.

Response: We have made the necessary corrections. Thank you very much for the comment.

2) There are typing errors in the manuscript that need to be corrected.

Response: We are sorry for the typos and the inconvenience caused. Thank you very much for this comment. We have made the necessary modifications.

3) The authors have discussed COVID-19 and its effects on the male reproductive system in the introduction section. However, there is no direct report available that shows that COVID-19 impacts the male reproductive system, and the ones that are available and authors have cited are indicative only. The authors should tone down the potential claims while writing the reviews in this section. Also, add more references to support your claims.

Response: Thank you for this comment. We have modified the introduction section and tried adding the latest references to improve the quality of the manuscript. The added references are as follows

Çayan S, UÄŸuz M, Saylam B, Akbay E. Effect of serum total testosterone and its relationship with other laboratory parameters on the prognosis of coronavirus disease 2019 (COVID-19) in SARS-CoV-2 infected male patients: a cohort study. Aging Male. 2020 Dec;23(5):1493-1503. doi: 10.1080/13685538.2020.1807930. Epub 2020 Sep 3. PMID: 32883151.

Kadihasanoglu M, Aktas S, Yardimci E, Aral H, Kadioglu A. SARS-CoV-2 Pneumonia Affects Male Reproductive Hormone Levels: A Prospective, Cohort Study. J Sex Med. 2021 Feb;18(2):256-264. doi: 10.1016/j.jsxm.2020.11.007. Epub 2020 Nov 27. PMID: 33468445; PMCID: PMC7691132.

Khalili MA, Leisegang K, Majzoub A, Finelli R, Panner Selvam MK, Henkel R, Mojgan M, Agarwal A. Male Fertility and the COVID-19 Pandemic: Systematic Review of the Literature. World J Mens Health. 2020 Oct;38(4):506-520. doi: 10.5534/wjmh.200134. Epub 2020 Aug 14. PMID: 32814369; PMCID: PMC7502312.

Selvam, M.K.P. and Sikka, S.C., 2022. Role of endocrine disruptors in male infertility and impact of COVID-19 on male reproduction. In Reproductive and Developmental Toxicology (pp. 1183-1194). Academic Press.

Selvaraj K, Ravichandran S, Krishnan S, Radhakrishnan RK, Manickam N, Kandasamy M. Testicular Atrophy and Hypothalamic Pathology in COVID-19: Possibility of the Incidence of Male Infertility and HPG Axis Abnormalities. Reprod Sci. 2021 Oct;28(10):2735-2742. doi: 10.1007/s43032-020-00441-x. Epub 2021 Jan 7. PMID: 33415647; PMCID: PMC7790483.

Sengupta P, Dutta S. COVID-19 and hypogonadism: secondary immune responses rule-over endocrine mechanisms. Hum Fertil (Camb). 2021 Jan 13:1-6. doi: 10.1080/14647273.2020.1867902. Epub ahead of print. PMID: 33439057.

Temiz MZ, Dincer MM, Hacibey I, Yazar RO, Celik C, Kucuk SH, Alkurt G, Doganay L, Yuruk E, Muslumanoglu AY. Investigation of SARS-CoV-2 in semen samples and the effects of COVID-19 on male sexual health by using semen analysis and serum male hormone profile: A cross-sectional, pilot study. Andrologia. 2021 Mar;53(2):e13912. doi: 10.1111/and.13912. Epub 2020 Nov 26. PMID: 33244788; PMCID: PMC7744846.

4) In section 3.1.3, two reports are cited that showed the role of SARS-CoV-2 in erectile dysfunction (53,55). Please describe the nature of the studies and the number of samples used in these papers. Also, rephrase line 181.

Response: Thank you very much for this comment. We have made the necessary modifications and have described the nature of the studies and the number of samples used in the papers, which are as follows

A pilot research describing the histological characteristics of penile tissue in subjects who healed from symptomatic COVID-19 infection and acquired significant erectile dysfunction. Patients with severe erectile dysfunction who underwent penile prosthesis surgery provided their penile tissue for this study. Two males with known COVID-19 infection histories and two without such histories provided samples. The findings of this study are the first to show that the COVID-19 virus can persist in the human penis for a long time after infection. This study adds to the growing evidence linking COVID-19 infection and erectile dysfunction by suggesting that extensive endothelial cell dysfunction may play a role.

Another study examined the incidence of erectile dysfunction among participants with a recorded diagnosis of COVID-19 and the relationship between COVID-19 and erectile dysfunction. This study analyzed data from the Sex@COVID online survey to get a sample of sexually active Italian males with SARS-CoV-2 infection. This study found preliminary evidence in a real-world population that erectile dysfunction is a risk factor for acquiring COVID-19 and may result from COVID-19.

5) In section 3.1.4, The role of oxidative stress is described in COVID-19 and its implications for male fertility. Oxidative stress is general in numerous infectious diseases. Please highlight the difference in covid 19 and other conditions that lead to male infertility.

Response: We have made the necessary modifications. Thank you for this comment.

6) Lines 186, 189, 257, 288, 291, 318, and 380 lack referencing. Please recheck all your references and update them with relevant studies (e.g., 91). Please cite only relevant papers and remove irrelevant references.

Response: We have made the necessary modifications. Thank you for this comment.

7) Remove line 336. If testosterone makes men more susceptible, young men should get more infections than older men. There are various factors that lead to the prognosis of COVID-19. For example, is it not common to have low testosterone in a diseased person due to a lack of an active lifestyle and other factors?

Response: We have removed the line for improving the quality of the manuscript

8) Rephrase line 370.

Response: We have removed the line to prevent any confusion and to improve the quality of the manuscript.

9) Please improve the discussion by complementing/supporting more profound and recent literature searches. 

Response: We have referred several recently published manuscripts and have improved the quality of the discussion section.

10) The manuscript could benefit from discussing the challenges and future prospects of the field. (optional)

Response: We have included the challenges and future prospects of the field. The same is as follows

Some of the most commonly used drugs for the treatment of COVID-19 are steroidal drugs, antiviral drugs, and antibiotics [160]. The potential reproductive harm of antiviral medications is frequently overlooked during COVID-19 therapy. Some of the most common antiviral drugs are ribavirin, Remdesivir, interferon, ivermectin, lopinavir, chloroquine, and ritonavir [161, 162]. Ribavirin, an antiviral drug, was found to impair testosterone levels and dysregulated spermatogenesis [163]. In a clinical study in rats, ribavirin was found to induce sperm abnormalities. This hazardous impact, however, was reversible. [39] Clinical trials revealed that ribavirin paired with interferon therapy caused male infertility by lowering sperm count [164, 165]. Remdesivir is another FDA-approved drug for COVID-19 treatments that haven’t reported any adverse effects on the male reproductive system [166]. Ritonavir has been shown to cause oxidative damage in the testis of rats, resulting in alterations in sperm characteristics and the animals' antioxidant state [167]. Glucocorticoids have been linked to germ-cell death via receptors on germ cells.  As a result, glucocorticoids are only indicated for a limited time in the treatment of COVID-19 individuals with rapid progression of imaging symptoms, progressive worsening of oxygenation markers, and significant activation of inflammatory responses in the body [168]. Various other studies focused on the role of antibiotics against COVID-19 and found that therapeutic doses of penicillin-G could lead to a spermatogenic arrest in rats after eight days of therapy [169]. More research is needed to understand the role of different biomolecules against male infertility followed by COVID-19 infection. Notably, we cannot claim with absolute confidence that the unique SARS-CoV-2 directly infects the testes, but it is possible, as with other viral infections, that this virus might wreak havoc on reproductive tissues, severely impacting the sexual health and fertility of males. Conversely, any other viral condition can temporarily diminish sperm count for several weeks or months. This makes it challenging to determine whether the decrease in sperm seen in the evaluated research is directly attributable to COVID-19. Regrettably, in most investigations, a semen study was not performed prior to SARS-CoV-2 infection; hence, the semen changes reported in COVID-19 patients may have been present prior to SARSCoV-2 infection [5]. Future studies on ACE2 and its effects on male genital tissues will necessitate investigation. Altogether, the results presented in this study suggested that, due to its high ACE2 expression, the testis is the organ in the male reproductive system that is most susceptible to SARS-CoV-2 infection. The role of the ACE2 receptors in spermatogonia, seminiferous tubules, Sertoli, Leydig, and prostate epithelial cells is uncertain [27]. There are different types of natural compound sources that have been found beneficial in the treatment of SARS-CoV-2 infection in male disorders. Table 2 summerizes the same.

11) Discuss natural products and medicinal compounds that could benefit male fertility after COVID-19. A tabular representational would be better for the readers.

Response: We have discussed and tabulated the natural and medicinal compounds that could benefit male fertility after COVID-19. Thank you for this comment.

Table 2: The potential role of natural compounds to benefit male fertility after COVID-19 infection.

Sources

Scientific name

Study type

Study model

Dose

Effects

Mechanism

Reference

Garlic

Allium sativum

In-vivo

Adult male albino rats of the Sprague–Dawley strain

30g

Modify spermatogenesis; spermicidal effect

↓plasma and intratesticular testosterone concentration

[170]

Ginger

Zingiber officinale

In-vivo

Japanese quail (Coturnix coturnix japonica)

50, 100 and 150 μl/kg bw

↓ reproductive cell lipid peroxidation, ↑ fertility

↓ impairment in sperm membrane and DNA

[171]

Grape

---------

In-vivo

Wistar rats

2 mg/kg bw

↑ Leydig cell percentage

↑ FSH, LH and testosterone production; prevent the spermatogenic disruption

[172]

Green tea

Camellia sinensis

In-vivo

Adult male mice

150 mg/kg bw

↓ apoptosis, testicular tissue changes, MDA; improve sperm parameters

↓ free radicals and expression of caspase-3;↑ testosterone levels

[173]

Micro-algae and algae

Atrhrospira platensis

In-vivo

Barki ram-lambs

---------

↑ total sperm output and total motile sperm

↑ anti-oxidative status and semen quality

[174]

Propolis

---------

In-vivo

Adult male Sprague Dawley rats

50 mg/kg

↓ MDA, GSSG levels; ↑ GSH, ATP levels

↑ sperm count, motility and validity; ↓ sperm abnormality

[175]

12) Include information in the discussion about preventive biomolecules available. (optional)

 Patel SKS, Lee JK, Kalia VC. Deploying Biomolecules as Anti-COVID-19 Agents. Indian J Microbiol. 2020 Sep;60(3):263-268. doi: 10.1007/s12088-020-00893-4. Epub 2020 Jun 9. PMID: 32647390; PMCID: PMC7282542.

Response: We have discussed the preventive biomolecules available. Thank you very much for this comment.

Reviewer 3 Report

Dear authors, greetings!

The manuscript "Insights into the scenario of SARS-CoV-2 infection in male reproductive toxicity" reviews information available on SARS-CoV-2 infection focusing on male reproductive system: erectile dysfunction, oxidative stress, vasculitis, azoospermia and cryptozoospermia, anti-sperm antibodies production, alterations in androgens level and damages in brain in prostate that can lead to infertility.

The manuscript was well designed and written. However, prior to publishing, small adjustments can be performed to improve its quality, optimizing understanding:

1) To present the meaning of RAAS system in the first time the term is used (Figure 1). The meaning is presented in line 128 and the term is used for the first time in line 84.

2) To add a figure to section 3.1.1 to illustrate RAAS system pathway. 3) To add a table summarizing the information presented regarding virus’ effect on different aspects related to male infertility.

English language and style are fine and only minor spell check required.

Author Response

Reviewer 3

The manuscript "Insights into the scenario of SARS-CoV-2 infection in male reproductive toxicity" reviews information available on SARS-CoV-2 infection focusing on male reproductive system: erectile dysfunction, oxidative stress, vasculitis, azoospermia and cryptozoospermia, anti-sperm antibodies production, alterations in androgens level and damages in brain in prostate that can lead to infertility.

The manuscript was well designed and written. However, prior to publishing, small adjustments can be performed to improve its quality, optimizing understanding:

Response: We are very much thankful to the reviewer for reviewing this manuscript and providing such valuable comments to improve the quality of the manuscript. We have again carefully revised the entire manuscript and have tried to address all the issues very carefully.

1) To present the meaning of RAAS system in the first time the term is used (Figure 1). The meaning is presented in line 128 and the term is used for the first time in line 84.

Response: We have modified the position of the figure in the manuscript to avoid confusion. Thank you for this comment.

2) To add a figure to section 3.1.1 to illustrate RAAS system pathway.

Response: We have added a figure to section 3.1.1 to illustrate the RAAS system pathway. The figure is as follows

Figure: This figure gives an overview of the RAAS system pathway.

3) To add a table summarizing the information presented regarding virus’ effect on different aspects related to male infertility.

Response: We have added a table summarizing the information presented regarding virus’ effect on different aspects related to male infertility.

Table 1: Effects of SARS-CoV-2 on various male reproductive parameters leading to male infertility.

Infertility issues

Cause

Factors effecting

Immune response

References

Erectile dysfunction

By endothelial dysfunction

Age, diabetes mellitus, dyslipidaemia, hypertension, cardiovascular disease, BMI/obesity/waist circumference

-

[144]

Spermatogonia

By immune or inflammatory reactions

increased body temperature

Increase in proinflammatory cytokines, chemokines, including IL-6, TNF-α, and MCP-1

[145]

Vasculitis

Thrombosis and thromboembolism

Infections

CD3+ and CD68+ leucocytes thrombosis, elevated seminal levels of IL-6, TNF-α, and MCP-1

[146]

Azoospermia

Decreased ACE2 Levels, Bilateral orchitis

Antibiotic or other drugs used for treatments

Dyregulation of RAAs components

[147, 148]

Testis damage

Increased ACE2 in testis

Infection severity

trigger an autoimmune response and autoantibody production

[103]

Prostate cancer

highly expresses TMPRSS2, ACE2

Compromised response to SARS-CoV-2-derived antigens

compromised responsiveness

[149]

English language and style are fine and only minor spell check required.

Response: We have made the necessary modifications. Thank you for this comment.

Round 2

Reviewer 2 Report

The manuscript has undergone significant improvements. However, it is recommended that the authors focus on improving the manuscript's flow to make it more cohesive for readers. The manuscript currently appears fragmented. 

Comments:

1) The writers have made changes to the document without highlighting everything. Please send us a new version with a list of all the changes made.

2) In line 44, "in dozens of nations," rephrase.

3) In line 45, virus's destructive power", rephrase.

4) In line 47, "viral action," rephrase.

5) In line 57, "the method of involvement", rephrase.

6) In line 66, due to this needs to be clarified.

7) In line 68 needs reference.

8) Improve the flow of the manuscript, which is challenging.

9) In line 89 no reference.

10) Ang II: given abbreviation first, then use a short form.

11) pro-moter. rewrite.

12) Chapter 3 has so many headings. It becomes difficult to follow this section. Arrange the heading in the review better. Some main headings are similar to the subheading.

13) Under 3.1.1 and 3.1.2, the authors give information about the basis for ACE2 and RAAS, ACE2 and TMPRSS2, but how it leads to male fertility is not explained.

14) In lines 163 and 166, the reference is missing.

15) In table 2, Important Chinese herbs  or Indian herbs are missing.

16) Discussion and future perspectives have abrut ending.

17) Remove the unnecessary details in SARS-CoV-2 in azoospermia and cryptozoospermia (3.1.6).

18 Remove the unnecessary lines from the manuscript without disturbing the flow of the manuscript.

Author Response

The manuscript has undergone significant improvements. However, it is recommended that the authors focus on improving the manuscript's flow to make it more cohesive for readers. The manuscript currently appears fragmented. 

We are very much thankful to the reviewer for reviewing this manuscript and providing such valuable comments to improve the quality of the manuscript. We have again carefully revised the entire manuscript and have tried to address all the issues very carefully.

Comments:

1) The writers have made changes to the document without highlighting everything. Please send us a new version with a list of all the changes made.

Response: We have edited the manuscript with the track changes feature. So, all the changes made can be easily found now. Thank you for this comment.

2) In line 44, "in dozens of nations," rephrase.

Response: We have rephrased the line. Thank you very much for this comment.

3) In line 45, virus's destructive power", rephrase.

Response: We have rephrased the line. Thank you very much for this comment.

4) In line 47, "viral action," rephrase.

Response: We have rephrased the line. Thank you very much for this comment.

5) In line 57, "the method of involvement", rephrase.

Response: We have rephrased the line. Thank you very much for this comment.

6) In line 66, due to this needs to be clarified.

Response: We have rephrased the line. Thank you very much for this comment.

7) In line 68 needs reference.

Response: We have added the reference. Thank you very much for this comment.

8) Improve the flow of the manuscript, which is challenging.

Response: We have made the necessary modifications. Thank you very much for this comment.

9) In line 89 no reference.

Response: We have added the reference. Thank you very much for this comment.

10) Ang II: given abbreviation first, then use a short form.

Response: We have made the necessary modifications. Thank you very much for this comment.

11) pro-moter. rewrite.

Response: We have made the necessary modifications. Thank you very much for this comment.

12) Chapter 3 has so many headings. It becomes difficult to follow this section. Arrange the heading in the review better. Some main headings are similar to the subheading.

Response: We have made the necessary modifications. We have arranged the headings properly. Thank you very much for this comment.

13) Under 3.1.1 and 3.1.2, the authors give information about the basis for ACE2 and RAAS, ACE2 and TMPRSS2, but how it leads to male fertility is not explained.

Response: We have made the necessary modifications. We have added the necessary details regarding ACE2, RAAS, ACE2, and TMPRSS2 to clarify their role in male infertility. Thank you very much for this comment.

14) In lines 163 and 166, the reference is missing.

Response: We have made the necessary modifications. We have added the required references. Thank you very much for this comment.

15) In table 2, Important Chinese herbs  or Indian herbs are missing.

Response: We have made the necessary modifications. We have added information regarding the essential Chinese herbs. Thank you very much for this comment.

16) Discussion and future perspectives have abrut ending.

Response: We have made the necessary modifications. We have improved the discussion and the future perspectives section to improve the quality of the manuscript.

17) Remove the unnecessary details in SARS-CoV-2 in azoospermia and cryptozoospermia (3.1.6).

Response: We have made the necessary modifications. We have removed the unnecessary details in SARS-CoV-2 in azoospermia and cryptozoospermia.

18 Remove the unnecessary lines from the manuscript without disturbing the flow of the manuscript.

 Response: We have made the necessary modifications. We have removed the unnecessary lines. Thank you for this comment.
